# AhR, PXR and CAR: From Xenobiotic Receptors to Metabolic Sensors

**DOI:** 10.3390/cells12232752

**Published:** 2023-11-30

**Authors:** Leonida Rakateli, Rosanna Huchzermeier, Emiel P. C. van der Vorst

**Affiliations:** 1Institute for Molecular Cardiovascular Research (IMCAR), RWTH Aachen University, 52074 Aachen, Germany; lerakateli@ukaachen.de (L.R.); rhuchzermeie@ukaachen.de (R.H.); 2Aachen-Maastricht Institute for CardioRenal Disease (AMICARE), RWTH Aachen University, 52074 Aachen, Germany; 3Institute for Cardiovascular Prevention (IPEK), Ludwig-Maximilians-University Munich, 80336 Munich, Germany; 4Interdisciplinary Center for Clinical Research (IZKF), RWTH Aachen University, 52074 Aachen, Germany

**Keywords:** lipid metabolism, cardiometabolic diseases, xenobiotic receptors, aryl hydrocarbon receptor, pregnane X receptor, constitutive androstane receptor

## Abstract

Traditionally, xenobiotic receptors are known for their role in chemical sensing and detoxification, as receptor activation regulates the expression of various key enzymes and receptors. However, recent studies have highlighted that xenobiotic receptors also play a key role in the regulation of lipid metabolism and therefore function also as metabolic sensors. Since dyslipidemia is a major risk factor for various cardiometabolic diseases, like atherosclerosis and non-alcoholic fatty liver disease, it is of major importance to understand the molecular mechanisms that are regulated by xenobiotic receptors. In this review, three major xenobiotic receptors will be discussed, being the aryl hydrocarbon receptor (AhR), pregnane X receptor (PXR) and the constitutive androstane receptor (CAR). Specifically, this review will focus on recent insights into the metabolic functions of these receptors, especially in the field of lipid metabolism and the associated dyslipidemia.

## 1. Introduction

Cardiometabolic diseases (CMDs), including atherosclerosis and non-alcoholic fatty liver disease, remain the leading cause of mortality worldwide [1]. Dyslipidemia is a fundamental and interlinked component of CMDs as it represents a major risk factor for their initiation and progression. The prevalence of dyslipidemia has been steadily rising in recent decades, in part due to changing lifestyles and dietary habits. Dyslipidemia encompasses a range of lipid abnormalities, such as elevated concentrations of low-density lipoprotein (LDL) and triglycerides (TG), as well as reduced levels of high-density lipoprotein cholesterol (HDL) [2]. Despite the effectiveness of lipid-lowering medications, such as statins, a significant residual risk for these diseases remains in patients [3]. The intricate molecular mechanisms of the transcriptional factors governing lipid homeostasis make it unlikely for a single agent targeting just one or two aspects of lipid metabolism to prevail. Therefore, a profound understanding of the genetic and transcriptional control of lipid metabolism at a molecular level is crucial, as it lays the groundwork for more precise and focused strategies in both prevention and intervention [4].

## 2. Lipid Metabolism

As the largest metabolic organ of the human body, the liver has a key role in maintaining lipid homeostasis in the body by controlling the uptake of lipids, lipid storage and lipid consumption [5,6]. A disruption in lipid metabolism can lead to abnormal levels of lipids and their metabolites in the blood or organs, resulting in hyperlipidemia [7]. Lipids include, among others, cholesterol, triglycerides and phospholipids, which are all hydrophobic molecules. Therefore, lipids need to bind to apolipoproteins in order to form lipoproteins, to allow their transport through the circulation to other organs and to be metabolized when necessary [8]. The classification of lipoproteins is based on their size and density, which is determined by the protein to lipid ratio. This ratio relies on the specific composition of lipids and apolipoproteins in each lipoprotein. The plasma lipoproteins include chylomicrons, very low-density lipoprotein (VLDL), intermediate-density lipoprotein (IDL), LDL and HDL [9].

In general, lipid metabolism is divided into two main pathways: the exogenous and endogenous pathways (Figure 1). The exogenous pathway starts with the dietary intake of lipids, whose components, cholesterol and triglycerides, are absorbed by the intestine. After absorption, cholesterol and triglycerides are re-esterified into chylomicrons, which are released into the intestinal lymph before entering the bloodstream. Upon reaching the capillary walls in adipose and skeletal muscle tissue, they interact with the enzyme lipoprotein lipase (LPL). This interaction results in the hydrolysis of the triglyceride core and the release of free fatty acids. These liberated free fatty acids can traverse the capillary endothelial cells and, for example, reach adipocytes and skeletal muscle cells, where they can be stored or used for energy through oxidation. The removal of the triglyceride core leads to the formation of remnant chylomicron particles, which are high in cholesterol esters and can be taken up by the LDL receptor (LDLR) into the liver [10]. 

The endogenous pathway starts in the liver, where triglycerides, produced by the liver using free fatty acids and carbohydrates as substrates, along with cholesterol esters, are transferred to apoB-100 for the synthesis of VLDL particles [10,11]. In addition to apoB-100, VLDL also contains other apolipoproteins, like apoC-II and apoE. After the transportation of VLDL particles to peripheral tissues, the triglyceride core is hydrolyzed by LPL and fatty acids are released. After the reduction of the VLDL core, a remnant particle called IDL is formed. IDL acquires cholesterol esters from HDL and subsequently binds with them, leading to the formation of LDL [12].

LDL has the ability to transport cholesterol to peripheral tissues and is later cleared via endocytosis, which is mediated by hepatic LDLR. The level of hepatic LDLR plays an important role in regulating plasma LDL levels since a decrease in hepatic LDLR results in increased plasma LDL levels. The regulation of LDLR in the liver primarily depends on the cholesterol levels within the hepatocytes. This process is controlled by sterol regulatory element binding proteins (SREBPs), which are transcription factors that control the expression of LDLR and other essential genes involved in cholesterol and fatty acid metabolism. If cellular cholesterol levels are increased, SREBPs remain inactive in the endoplasmic reticulum and the synthesis of LDLRs stops. Conversely, when cellular cholesterol levels decrease, SREBPs are activated and stimulate LDLR synthesis to increase the uptake of cholesterol [11]. 

While VLDL, IDL and LDL are responsible for triglyceride and cholesterol transport from the liver to peripheral tissues, and HDL particles are responsible for reverse cholesterol transport from cells in the peripheral tissues to the liver (Figure 1) [9]. The maturation of HDL starts with the synthesis of apoA-I, the main structural protein found in HDL, by the liver and intestines. Following its secretion, the newly synthesized apoA-I (pre-beta HDL) subsequently acquires cholesterol and phospholipids from hepatocytes and enterocytes, which is mediated by ATP-binding cassette transporter A1 (ABCA1). In order to form mature HDL with a core of cholesterol esters, the free cholesterol by HDL must be esterified via the enzyme lecithin cholesterol acyl transferase (LCAT). These fully mature HDL particles have the ability to transport cholesterol from the peripheral tissues to the liver, facilitated by the scavenger receptor B-I (SR-BI) and cluster of differentiation 36 (CD36), or exchange cholesterol and triglycerides between apoB-containing particles VLDL and LDL, which is mediated by cholesteryl ester transfer protein (CETP) [4,11]. 

## 3. Xenobiotic Receptors

Over the last few years, there has been growing interest in a specific group of receptors called nuclear receptors, due to their significant involvement in lipid metabolism. Nuclear receptors function as transcription factors that serve as sensors of extracellular and intracellular signals. They play pivotal roles in governing biological development, cellular differentiation, the maintenance of metabolic balance and protection against xenobiotic-induced stress [13]. The term xenobiotic refers to chemical substances that are not naturally occurring in the life of an organism. Examples of xenobiotics include pesticides, drugs, food additives and environmental pollutants [14]. Nuclear receptors can be broadly categorized into two main classes, distinguished by their ligand-binding specificity: endocrine and orphan nuclear receptors. Endocrine receptors, such as the estrogen receptor, exhibit high-affinity binding to specific endogenous ligands, including hormones and steroids, that are naturally found in low physiological concentrations. In contrast, orphan nuclear receptors typically lack identified high-affinity endogenous ligands and instead are activated by abundant and low-affinity metabolites or xenobiotics [15]. Within this group, several orphan nuclear receptors have the ability to bind to a broad spectrum of both endogenous compounds and xenobiotics. They play a crucial role in orchestrating cellular reactions to harmful compounds and their metabolites. This specific group of orphan nuclear receptors, referred to as xenobiotic receptors, have been recognized as important sensors of toxic byproducts resulting from the breakdown of both endogenous and exogenous chemicals, as well as key mediators in coordinating subsequent toxic and protective responses [16,17]. Two prominent members of the nuclear receptor family are the pregnane X receptor (PXR) and the constitutive androstane receptor (CAR), which function as xenobiotic receptors. 

Xenobiotic receptors control the transcriptional activation of genes that encode phase I and II drug-metabolizing enzymes as well as uptake and efflux transporters [17]. The detoxification and elimination of xenobiotics involve a coordinated interplay among phase I cytochrome P450 (CYP) enzymes, phase II conjugating enzymes and drug transporters [18]. These CYP enzymes facilitate the monooxygenase reactions of lipophilic compounds by utilizing the reducing power provided by nicotinamide adenine dinucleotide phosphate hydrogen (NADPH) P450 oxidoreductase [19]. Phase II enzymes encompass various transferases, including sulfotransferase (SULT) and UDP-glucuronosyltransferase (UGT), which attach polar functional groups to xenobiotics [20]. The final step involves the participation of ABC transporter proteins and members of the solute carrier family, which facilitate the excretion process [21]. Three major xenobiotic receptors are the aryl hydrocarbon receptor (AhR), PXR and CAR, which are all predominantly found in the liver and intestines, where their target genes are also situated. In contrast to CAR and PXR, AhR does not belong to the family of nuclear receptors but is a ligand-activated transcription factor that belongs to the Per-Arnt-Sim (PAS) superfamily of proteins, exhibiting similar properties to nuclear receptors. Together, these xenobiotic receptors collectively govern a wide array of target genes, effectively coordinating a hepatoprotective system in response to detrimental environmental factors [13]. 

## 4. Xenobiotic Receptors in Lipid Metabolism

The xenobiotic receptors AhR, PXR and CAR have historically been recognized for their involvement in regulating the detoxification and metabolism of foreign compounds. However, recent research has unveiled their emerging functions beyond xenobiotic metabolism. It has been shown that these xenobiotic receptors influence many cellular processes outside of their traditional realms, such as the metabolic homeostasis of lipids, demonstrating a novel role as metabolic sensors [22]. The term xenosensors, which initially described the role of PXR and CAR, is now being changed to the term metabolic sensors, as more and more studies link their activity to metabolic pathways such as glucose or lipid metabolism [4]. Disrupted lipid signaling mediated by metabolic receptors can perturb homeostatic pathways, consequently playing a role in the development of CMDs, such as obesity, hyperlipidemia and atherosclerosis. Given their integral participation in metabolic processes, these receptors emerge as intriguing targets for therapeutic interventions in the treatment of CMDs [4]. Therefore, this review discusses the role of the xenobiotic receptors AhR, PXR and CAR in lipid metabolism. 

### 4.1. Role of AhR in Lipid Metabolism

The AhR protein is present in numerous tissues, exhibiting the highest mRNA and protein concentrations in the lungs, liver, kidney and placenta. The activation of AhR signaling is acknowledged as the primary molecular defense mechanism in the body when exposed to environmental toxicants [22]. AhR signaling starts in the cytoplasm of cells, where, in its inactive state, AhR is bound to heat shock protein 90 (Hsp90), co-chaperone p23, protein kinase SRC and AhR interacting protein (AIP) (Figure 2; left panel). In the canonical pathway, this complex releases AIP upon ligand-based receptor activation and translocates into the nucleus, where the cofactors dissociate and AhR heterodimerizes with the AhR nuclear translocator (ARNT) [23]. The AhR–ARNT complex then binds to a specific DNA recognition sequence (5′-TNGCGTG-3′), which results in the transcription of several downstream genes, including cytochrome P450 family 1 subfamily A member 1 (*CYP1A1*) and cytochrome P450 family 1 subfamily B member 1 (*CYP1B1*), both involved in the metabolism of endogenous hormones and xenobiotics. The transcription of the aryl hydrocarbon receptor repressor (*AHRR*) acts as a negative regulatory mechanism [23,24,25].

AhR was initially identified as a regulator of xenobiotic metabolism, specifically of polycyclic aromatic hydrocarbons (PAHs), such as 2,3,7,8-Tetrachlorodibenzo-p-dioxin (TCDD) [26,27]. Previous research had already established a link between the AhR agonist TCDD and dyslipidemia, as it had been reported that TCDD treatment resulted in hepatic steatosis in mice and rats [28,29,30]. Additionally, exposure to dioxin in human populations has been correlated with the higher occurrence of non-alcoholic fatty liver disease (NAFLD) [31]. The hepatic steatosis observed in TCDD-treated rats could potentially be attributed to an elevated rate of de novo fatty acid synthesis, a decrease in fatty acid oxidation and an extended half-life of liver lipid components [32,33,34]. A study by Ambolet-Camoit et al. revealed that lipid deposition can be specifically induced by several ligands of AhR, such as TCDD and α-endosulfan [35]. Nevertheless, certain studies have presented an alternative perspective, indicating reduced hepatic fatty acid synthesis in response to TCDD treatment in both animal and primary human hepatocytes. This reduction was associated with the diminished expression of crucial lipogenic genes, including fatty acid synthase (FASN), stearyol-CoA desaturaturase-1 (SCD-1) and acetyl-CoA carboxylase-1 (ACC-1) [36,37,38]. Interestingly, however, one of the earliest studies, highlighting AhR as a potential therapeutic target in lipid metabolism, demonstrated that the regulatory effect of the cholesterol synthesis pathway by AhR is independent of its dioxin response element [39].

In a recent study, it was demonstrated that mice with the liver-specific overexpression of a constitutively active AhR (CA-AhR) developed spontaneous steatosis without inducing widespread hepatotoxicity [40]. The steatosis observed in CA-AhR transgenic mice was characterized by the enhanced uptake of fatty acids, increased hepatic oxidative stress, the elevated expression of CD36 and a reduction in VLDL triglyceride secretion [40]. The fatty acid translocase CD36 was recognized as a newly discovered transcriptional target gene of AhR, responsible for mediating the development of steatosis [40]. Other studies showed that both whole-body AhR deficiency and the use of platelet-derived growth factor receptor alpha (Pdgfrα)-Cre-mediated AhR knockout in preadipocyte lineages and other cell types shielded mice from diet-induced obesity and metabolic disorders by promoting higher energy expenditure [41,42]. However, liver-specific AhR knockout in mice resulted in severe hepatic lipotoxicity, demonstrated by severe steatosis, inflammation and injury in the livers of KO mice. The de novo lipogenesis activity was significantly upregulated in these liver-specific AhR KO mice, which was shown by the increased expression of genes involved in lipogenesis, such as the sterol response element-binding protein-1c (Srebp-1c). Moreover, the deletion of the AhR gene in the liver attenuated the expression of the suppressor of cytokine signal (Socs3), which is a negative inflammatory factor. The study revealed that the hepatic lipotoxicity was triggered by the transcriptional regulation of Socs3 expression by AhR, establishing Socs3 as a novel transcriptional target of AhR [43]. These findings imply that the influence of AhR on lipid metabolism could vary depending on the specific ligands and the location of the receptor.

Using CA-AhR transgenic mice, recent research demonstrated that AhR activation sensitizes mice to non-alcoholic steatohepatitis (NASH), marked by inflammation and progressive fibrosis, especially when induced by a methionine- and choline-deficient diet (MCD) [44]. This heightened sensitivity was linked to reduced superoxide dismutase 2 (SOD2) activity and increased mitochondrial reactive oxygen species (ROS) production in the liver due to the AhR-mediated inhibition of the mitochondrial sirtuin deacetylase SIRT3 [44]. Interestingly, AhR can also have an antioxidative role, as it induces nuclear factor erythroid 2 related factor 2 (Nrf2) and other cytoprotective genes, suggesting its complex role in liver oxidative stress regulation [45,46].

Apart from its involvement in NAFLD and NASH, AhR activation is also associated with dyslipidemia and atherosclerosis. For example, workers and community residents exposed to TCDD exhibited lipid abnormalities, and a follow-up study on former TCDD-exposed workers revealed a link between TCDD exposure and atherosclerotic plaques and ischemic heart disease [47]. This was supported by an animal study, in which *Apoe*^−/−^ mice were exposed for 12 weeks to the AhR agonist TCDD, showing increased atherosclerotic lesions compared to untreated control *Apoe*^−/−^ mice [48]. The expression of several inflammatory marker genes was upregulated in these mice, including matrix metalloproteinase-12 (MMP-12), a parameter for activated macrophages. In particular, the chemokine receptor C-X-C motif chemokine receptor 2 (CXCR2) appeared to have a significant role in AhR signaling, as the CXCR2 activation induced by AhR seemed to enhance foam cell formation and contribute to the development of atherosclerotic lesions in TCDD-treated *Apoe*^−/−^ mice [48]. Moreover, other studies showed that TCDD induced dyslipidemia, which was linked to the decreased activity of LPL in adipose tissue, impairing triglyceride hydrolysis and cellular uptake [49,50,51,52]. Additionally, AhR deletion has demonstrated protective effects against high-fat-diet-induced dyslipidemia and vascular dysfunction, potentially by enhancing endothelial nitric oxide synthase/nitric oxide (eNOS/NO) signaling [53].

All in all, these results clearly demonstrate that AhR plays an important role in lipid metabolism, driving dyslipidemia and thereby exerting a detrimental effect on related CMDs (Figure 3).

### 4.2. Role of PXR in Lipid Metabolism 

Another master regulator of xenobiotic metabolism is PXR (also known as steroid and xenobiotic receptor [SXR]; NR1I2) [17,54,55,56]. Similar to the other xenobiotic receptors, PXR also induces the expression of genes that are required for xenobiotic metabolism, like CYP enzymes, conjugating enzymes and ABC family transporters [57,58].

The subcellular location of PXR is still a matter of debate. Several studies in murine cells demonstrate that PXR, in its transcriptionally inactive state, is present in the cytoplasm, where it forms a complex with Hsp90 and cytoplasmic CAR retention protein (CCRP), and only upon activation does it dissociate from its complex and translocate to the nucleus [59,60]. However, it has also been reported in human cells that PXR is localized only in the nucleus, independent of the presence or absence of a ligand [61,62]. A potential explanation behind this discrepancy may be the species difference, although it is clear that the nuclear localization of PXR is important for its activation. 

Unbound PXR in the nucleus forms a complex with the corepressors nuclear receptor corepressor 1 (NcoR1) and silencing mediator of retinoid and thyroid receptors (SMRT), and it also dissociates upon ligand stimulation [63]. Independent of the origin of the activated PXR, inside the nucleus, the coactivators steroid receptor coactivator 1 (SRC1) and glucocorticoid receptor interacting protein 1 (GRIP1) are recruited to activated PXR, initiating the heterodimerization of PXR with retinoid X receptor (RXR), followed by the induction of target gene expression by binding to a PXR response element [63] (Figure 2; middle panel). Additionally, many transcriptional cofactors have been shown to regulate PXR activity, such as members of the p160 family, like steroid receptor coactivators, as well as peroxisome proliferator-activated receptor (PPAR)-α [64].

In recent years, several studies have highlighted a pivotal endobiotic function of PXR in the regulation of lipid metabolism [65,66,67]. For example, PXR has been associated with NAFLD, as it could be shown that the expression of a key PXR target gene, Cyp3a11, increased significantly during NAFLD progression [68]. Although this only shows an associative relationship, other studies have also proven causality as transgenic mice with constitutively activated PXR showed a marked increase in hepatic steatosis, particularly due to hepatic triglyceride accumulation [69]. In line with this, the treatment of mice with a PXR agonist, rifampicin, also elicited steatosis [70]. Furthermore, mice lacking PXR demonstrated reduced hepatic steatosis [71], strengthening the notion that PXR plays a key role in hepatic lipid accumulation. However, it is important to note that mice also develop systemic hypertriglyceridemia upon PXR activation [72,73]. Several human-focused studies also corroborated these findings, as, for example, the activation of PXR in human hepatocytes increased cellular lipid accumulation [74]. Another study interestingly found that both the activation and the inhibition of PXR resulted in increased lipogenesis via divergent mechanisms [75]. Furthermore, a case–control association study of 290 individuals demonstrated that PXR polymorphisms were associated with disease severity in NAFLD, as well as increased alanine aminotransferase (ALT) levels, which is a surrogate marker of severe liver injury [76]. Strikingly, PXR polymorphisms are also reported to increase the risk of total mortality in subjects with NAFLD, but not in subjects without NAFLD [77].

This hepatic lipid accumulation upon PXR activation is likely the combined effect of various processes that are influenced. For example, the observed steatosis coincided with the increased expression of the free fatty acid transporter CD36 [69]. Interestingly, promoter analyses demonstrated that CD36 is a direct transcriptional target of PXR [69]. Additionally, PPARγ was also shown to be a direct target of PXR and is a positive regulator of CD36 [78]. These results could also be validated in another mouse model in which PXR was activated by feeding of the PXR agonist pregnenolone 16α-carbonitrile (PCN) [79]. Besides these effects on lipid uptake, PXR has also been implicated in lipogenesis, by, for example, inducing the thyroid hormone-responsive spot 14 protein (S14) involved in lipogenesis [80], lipin-1, known to increase triglyceride synthesis [81], or the solute carrier family 13 member 5 (SLC13A5), which plays an important role in fatty acid and cholesterol synthesis [82]. Furthermore, PXR is also involved in the regulation of β-oxidation and ketogenesis, by, for example, suppressing the rate-limiting enzymes carnitine palmitoyltransferase 1 (CPT1a) and 3-hydroxy-3-methylglutarate-CoA synthase (HMGCS2) [72]. This suppression occurred in an indirect manner, as PXR directly bound to forkhead box A2 (FoxA2), thereby preventing the binding to and thus activation of its target genes, like these enzymes [72]. The exact contribution of each of these mechanisms in the PXR-mediated effects remains an active field of research, although it is clear that PXR has a major influence on hepatic lipid levels.

Besides these effects on tissue lipid levels, PXR also plays a key role in lipoprotein homeostasis. In mouse models, the activation of PXR by PCN resulted in significantly increased total circulating cholesterol levels and increased levels of the atherogenic lipoproteins VLDL and LDL [79,83], while it decreased HDL cholesterol levels [84]. These effects of PXR on HDL metabolism are mediated by a multitude of effects as the receptor activation decreased the expression of genes involved in HDL synthesis (Abca1, Apoa1), maturation (Lcat, Pltp) and clearance (Sr-b1) [84,85]. However, other studies also demonstrated contradicting outcomes as the activation of PXR had stimulating effects on ApoA-I as well as HDL levels [86,87]. Although the exact reasons for this discrepancy remain to be elucidated, it is tempting to speculate that the use of different agonists may be the primary cause as these might induce divergent signaling cascades.

Interestingly, a recent study identified, in mice and humans, that PXR activation induced an increase in circulating proprotein convertase subtilisin/kexin type 9 (PCSK9) levels, which is a negative regulator of hepatic LDL uptake, which also coincided with increased circulating LDL levels [88]. Mechanistic studies identified that SREBP2 most likely plays an important regulatory role in this phenomenon [88]. However, it remains to be determined whether the regulation of PCSK9 and SREBP2 is dependent on the used mode of receptor activation.

Overall, it is clear that PXR has a major influence on lipid metabolism, by manipulating various processes. This notion has recently also been highlighted by an adverse outcome pathway framework [89]. Besides promoting the development of NAFLD, the activation of PXR by PCN has also been demonstrated to enhance atherosclerosis formation in *Apoe*^−/−^ mice, which is in line with its observed effects on the atherogenic VLDL and LDL levels [79]. Another model showed similar effects as the exposure of *Ldlr*^−/−^ mice to dicyclohexyl phthalate (DCHP) significantly increased atherosclerosis formation [90]. This effect was dependent on macrophage-expressed PXR, as mice lacking myeloid PXR were not affected by DCHP exposure [90]. The importance of myeloid PXR was also confirmed in a model in which no specific ligand treatment was performed, demonstrating that mice with a myeloid-specific deficiency of PXR had decreased atherosclerosis formation, such as in *Ldlr*^−/−^ mice [91]. 

However, it is important to keep in mind that most observations are only pre-clinical and should still be validated in a clinical setting. The importance of this is demonstrated by the study of Poulton et al., showing, in a randomized clinical trial, that sulforaphane does not have an effect on PXR, in contrast to clear pre-clinical data indicating an antagonistic function [92]. Further efforts for clinical validation have already been performed focusing on the PXR agonist rifampicin. Although no effects were observed on incretin hormone secretion [93], rifampin increases blood pressure and stimulates plasma renin activity in humans [94]. According to ClinicalTrials.gov, there are unfortunately no other clinical trials currently ongoing focusing on PXR targeting.

Combined, these results demonstrate that PXR has a major role in lipid metabolism and thereby also in various lipid-related CMDs (Figure 3).

### 4.3. Role of CAR in Lipid Metabolism 

CAR is also known as nuclear receptor NR113. Together with PXR, it is a member of the superfamily of nuclear receptors [95]. A variety of chemicals, such as phenobarbital, androstenol and clotrimazole, are potent ligands of CAR. Recent studies, however, suggest that the ligands of CAR are species-dependent, such as 1,4-bis-[2-(3,5-dichloropyridyloxy)]benzene and 3,3′,5,5′-tetrachloro-1,4-bis(pyridyloxy)benzene (TCPOBOP) only activate CAR in mice and not humans, while (E)-6-(4-chlorophenyl)imidazo[2,1-b]thiazole-5-carbaldehyde O-(3,4-dichlorobenzyl) oxime (CITCO) is a potent CAR ligand only in humans [96]. Some of the CAR ligands also activate its sister receptor PXR. Phenobarbitone and phenytoin, for example, activate both, while CITCO activates CAR more than PXR [97,98]. However, the outcome of the respective activation can vary between CAR and PXR. Phenytoin, for example, induces Cyp2b10 expression via CAR, whereas the Cyp3A4 induction by phenytoin is regulated via PXR activation [99].

CAR is located in the cytoplasm, where it is constitutively active [95]. The receptor is, in its inactive state, bound to co-chaperones such as Hsp90 and CCRP [95] (Figure 2; right panel). It can be activated via two pathways, a direct, ligand-based pathway and an indirect, ligand-independent pathway [95]. In the ligand-dependent activation pathway, the ligands directly bind to the phosphoprotein CAR in the cytoplasm. Thereby, the protein phosphatase 2A (PP2A) is recruited, which dephosphorylates CAR and thereby provokes the dissociation of CAR from its co-chaperones. This leads to its translocation into the nucleus, where it heterodimerizes with RXR. The heterodimer binds to the PB-responsive element module (PBREM), which alters the gene transcription in either an activating or inhibiting way [95]. The main target genes include phase I and phase II enzymes and drug reporters. The target genes mainly control cell proliferation and apoptosis, with double functions in drug metabolism and xenobiotic response [95].

The non-ligand-based or indirect activation of CAR takes places via the epidermal growth factor receptor (EGFR). If no ligand is present, EGF will bind to the EGFR, thereby provoking a signaling cascade via the activation of mitogen-activated protein kinase kinase (MEK) and src-kinase [100]. The activation of MEK results in the activation of the extracellular-signal-regulated kinase (ERK). Subsequently, this kinase inhibits the translocation of CAR into the nucleus. The receptor for activated C kinase (RACK-1) mediates the recruitment of PP2A to phosphorylate CAR and thereby enable the dissociation of its co-chaperones. The kinase src, which is active under EGFR signaling, inhibits RACK-1, thereby preventing its interaction with CAR. However, CAR activators can compete with EGF for binding to EGFR. Phenobarbital, for example, negatively regulates the MEK/ERK/src kinase activity and thereby supports CAR signaling. The resulting inhibition of the src kinase allows RACK-1 signaling, thereby permitting PP2A recruitment for the nuclear translocation of CAR [100]. CAR, in contrast to PXR, shows higher basal activity to activate target genes without ligands. Although CAR and PXR in particular have several overlapping genes, they also regulate distinctive sets of genes [101,102,103].

The liver is the main metabolic organ for lipid and glucose homeostasis [22]. The accumulation of bile acids can induce hepatotoxicity and thereby impact lipid homeostasis. The activation of CAR is preventive against this toxicity by downregulating the solute carrier organic anion transport protein 1a1 (OATP1A1) [104,105,106]. Additionally, the mRNA expression of bile acid efflux transporters such as OATP1A4 and multidrug-resistance-associated proteins (MRPs) is increased upon CAR activation, whereas uptake transporters such as OATP1B3 and OAT2 exhibit decreased mRNA expression upon CAR activation. Since bile acids play an important role in the absorption process of lipids [107], particularly of LDL in the intestine and the liver, the activation of CAR reduces systemic LDL levels [108,109,110,111].

The activation of CAR generally seems to be hepatoprotective, as CAR-deficient mice exhibited higher serum levels of ALT, which is strongly related to lipid accumulation in the liver [112], as well as increased necrosis in the liver [113]. In another mouse study by Baskin-Bey et al., the treatment of mice with the CAR agonist TCPOBOP resulted in reduced liver injury mediated by hindered hepatocyte apoptosis via reduced Bak and Bax and increased myeloid cell leukemia factor 1 expression [114]. Pavek and colleagues confirmed the lipid-reductive effect in wild-type mice upon TCPOBOP-dependent CAR activation. However, in their humanized CAR mouse model, TCPOBOP treatment led to increased serum and liver lipid levels [115]. Although TCPOBOP in the humanized mice did not seem to enhance CAR activation itself, the basal constitutive activity remained high [115]. In contrast to the hepatoprotective role of CAR, in a NASH mouse model, Car-/- mice did not show altered lipid levels but a decreased level of serum ALT and decreased lipid peroxidation in the liver, suggesting a rather detrimental role of CAR in lipid metabolism [116]. Furthermore, a study by Xie et al. unraveled a functional cross-talk between CAR and the nuclear receptor liver X receptor (LXR), which is a sterol receptor that exhibits lipogenic properties by supporting hepatic cholesterol catabolism [117]. Interestingly, the expression of LXR is increased due to the loss of CAR, thereby resulting in increased liver lipid levels [118].

Besides the indirect influence of CAR activation on lipid metabolism, many recent studies suggest that CAR activators such as drugs and environmental toxins, as well as endogenous metabolites, can also directly influence the lipid metabolism in the liver [22]. For example, in obese mice, the activation of CAR seemed to inhibit lipogenesis by lowering the mRNA expression of lipogenesis-associated genes such as Srebp-1c, Acc-1, Fas and Scd-1 [119,120]. Furthermore, the anti-lipogenic protein Insig-1, which blocks Srebp-1, is concurrently activated upon CAR activation [121]. In line with this, under metabolic stress conditions, mice lacking CAR exhibited higher serum triglycerides than mice with active CAR. This preventive function of CAR activity was associated with lower PPARα, Cyp4a14, carnitine palmitoyltransferase 1 α (CTP1α), cytosolic acyl-CoA thioesterase (CTE) and adiponectin expression in the liver, lowering hepatic fatty acid oxidation [122,123]. Consistent with these results, Masson et al. showed in an *Apoe*^−/−^ and *Ldlr*^−/−^ mouse model that treatment with the CAR agonist TCPOBOP resulted in decreased plasma and liver cholesterol, triglyceride and LDL levels, associated with limited VLDL production [124,125]. As a consequence, the atherosclerotic lesions in CAR-activated mice were also smaller than in untreated mice. Additionally, the reverse cholesterol transport seems to be affected by CAR activation, as HDL levels were increased upon treatment [125]. However, these results conflict with the findings of Assem et al., who suggested that TCPOBOP-induced CAR activation led to a decrease in plasma HDL cholesterol and apoA1 levels in a transgenic HFD mouse model. However, a loss of CAR activity does not seem to alter the HDL levels [126]. Furthermore, in a clinical setting, several interactions between CAR and lipids have been observed. For example, in a clinical study by El Hady et al., epileptic children treated with anti-epileptic drugs including phenytoin, a potent CAR activator, showed evaluated plasma lipid levels. These children showed elevated LDL, triglyceride and total cholesterol levels, while HDL levels seemed to be unaffected, increasing the risk for cardiovascular events in treated children, possibly mediated by the xenobiotic response [127]. Phenytoin induces CYP2B6 and CYP3A4 expression regulated mainly via CAR; however, phenytoin is also a PXR activator [99,128,129]. In other human studies, and in sharp contrast to the previously described study, treatment with phenytoin increased plasma HDL cholesterol levels, whereas the total cholesterol, triglyceride and LDL cholesterol levels remained unaffected under phenytoin treatment [130,131]. Therefore, the exact effects of CAR on lipoproteins remain an active field of research.

Since CAR seems to be involved in the regulation of lipid metabolism, it might be a plausible therapeutic target in the field of CMDs. However, since there are still various inconclusive results and inconsistencies between species, further research on the role of CAR in lipid metabolism is required.

## 5. Conclusions

The evolving understanding of the major xenobiotic receptors AhR, PXR and CAR extends far beyond their established roles in detoxification and xenobiotic metabolism. Recent investigations have unraveled their newfound significance as influential regulators of diverse cellular processes, particularly in lipid metabolism, thereby positioning them as essential metabolic sensors. Disruptions in lipid signaling mediated by these receptors can substantially disrupt the physiological equilibrium, ultimately contributing to the development of cardiometabolic disorders like hyperlipidemia, NAFLD and atherosclerosis. Consequently, the integration of these receptors into the intricate network of metabolic pathways underscores their potential as promising targets for future therapeutic advancements aimed at addressing and mitigating CMDs.

In spite of all this recent progress, a number of important questions demand increased attention to attain a comprehensive understanding of the impact of AhR, PXR and CAR on human health. Primarily, inconsistent effects of xenobiotic receptors were observed across different species. Thereby, the application of human studies will be essential to enhance our understanding of the relevance of xenobiotic receptors in lipid metabolism and metabolic diseases within the context of human physiology. Secondly, the xenobiotic receptors exhibited varying effects not only across different species but also within distinct tissues and organs of the same organism. In order to understand the multifaceted role of the xenobiotic receptors, future studies need to further investigate tissue or organ specificity, as knockout at the systemic level leads to general and imprecise conclusions.

## Figures and Tables

**Figure 1 cells-12-02752-f001:**
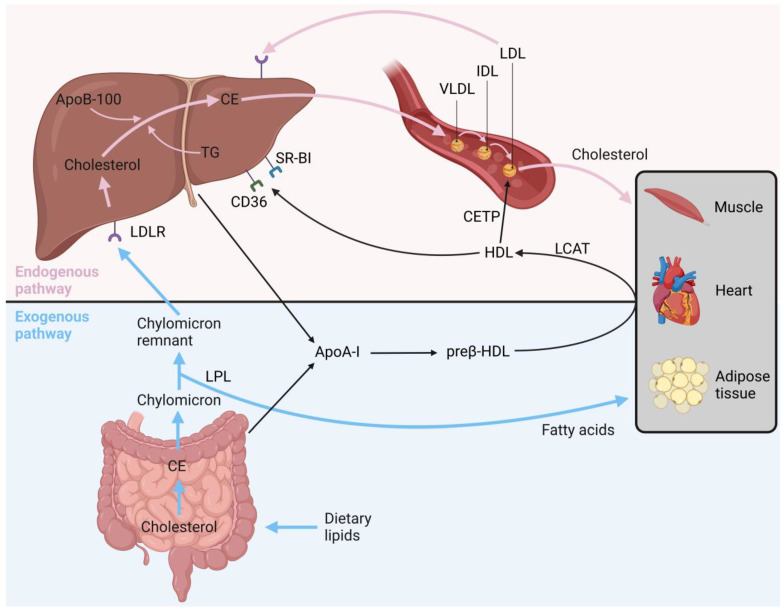
**Main pathways of lipid metabolism.** The exogenous pathway begins with the transport of ingested dietary lipids from the intestine to, for example, muscles and adipose tissue via chylomicrons, and the resulting chylomicron remnants transport the remaining lipids to the liver. The endogenous pathway starts in the liver with the synthesis of VLDL by conveying triglycerides and cholesterol esters in the endoplasmic reticulum to newly synthesized apoB-100 to form the core of VLDL. As VLDL circulates through the bloodstream, triglycerides are removed from VLDL, making it denser and more cholesterol-rich, eventually transforming into LDL. LDL primarily carries cholesterol to peripheral tissues, such as muscle, heart or adipose tissue, while HDL is responsible for reverse cholesterol transport, where excess cholesterol is transported from peripheral tissues back to the liver. ApoA-I: apolipoprotein A-I; ApoB-100: apolipoprotein B-100; CD36: cluster of differentiation 36; CE: cholesterol esters; CETP: cholesteryl ester transfer protein; HDL: high-density lipoprotein; IDL: intermediate-density lipoprotein; LCAT: lecithin cholesterol acyl transferase; LDL: low-density lipoprotein; LDLR: low-density lipoprotein receptor; LPL: lipoprotein lipase; SR-BI: scavenger receptor B-I; TG: triglyceride; VLDL: very low-density lipoprotein. Figure created with BioRender.com.

**Figure 2 cells-12-02752-f002:**
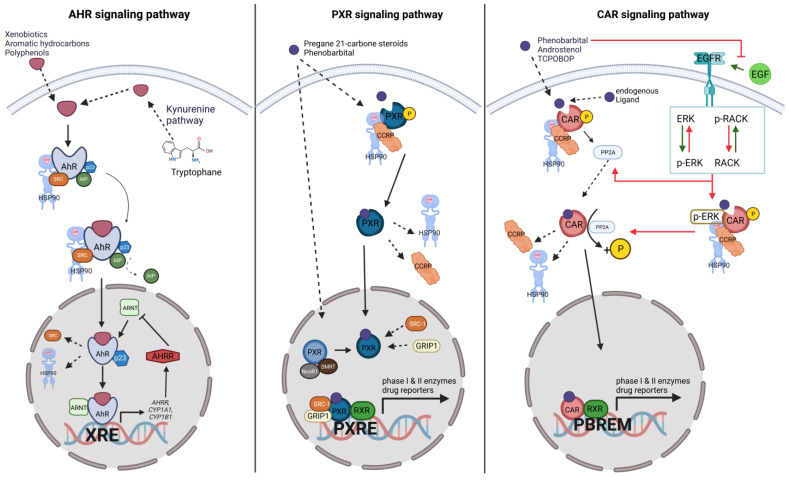
**Mechanism of xenobiotic receptor activation.** (**Left panel**): In the cytoplasm, AhR is bound by Hsp90, AIP, the co-chaperone p23 and protein kinase SRC. After ligand binding, with either exogenous ligands or endogenous ligands such as tryptophane metabolites, the AhR complex releases AIP and translocates to the nucleus, where the rest of the cofactors dissociate from AhR, enabling ARNT to heterodimerize with AhR. The AhR–ARNT complex then binds to the DNA and regulates the gene expression of several genes, such as CYP1A1, CYP1B1 or AHRR. AHRR can negatively regulate AhR by inhibiting the AhR–ARNT complex. (**Middle panel**): Mechanism of PXR activation. In the cytoplasm, PXR forms a complex with Hsp90 and CCRP, while non-activated PXR in the nucleus is bound to NcoR1 and SMRT. Upon receptor activation, PXR dissociates from its complex and translocates into the nucleus, where it binds to SRC-1 and GRIP1. Together with these coactivators, PXR heterodimerizes with RXR and binds to a PXR response element to induce target gene expression. (**Right panel**): CAR signaling. CAR is located in the cytoplasm, where it is bound to CCR and HSP90 in its inactive state. In the direct activation pathway, ligand binding activates PP2A, which dephosphorylates CAR, resulting in the dissociation of its co-chaperones. The activated CAR translocates into the nucleus, where it heterodimerizes with RXR and binds to the PBREM to induce target gene expression. In the indirect pathway (highlighted with green and red arrows), specific ligands of the Car can inhibit the binding of EGF to the EGFR. This results in the dephosphorylation of p-ERK and simultaneously the dephosphorylation of RACK. RACK activates PP2A and the dephosphorylated ERK dissociates from the CAR complex, enabling its activation by PP2A. AhR: aryl hydrocarbon receptor; AHRR: aryl hydrocarbon receptor repressor; AIP: AhR interacting protein; ARNT: aryl hydrocarbon receptor nuclear translocator; CAR: constitutive androstane receptor; CCRP: CAR cytoplasmic retention protein; CYP1A1: cytochrome P450 family 1 subfamily A member 1; CYP1B1: cytochrome P450 family 1 subfamily B member 1; EGF: epidermal growth factor; EGFR: EGF receptor; ERK: extracellular signal-regulated kinase, p-ERK: phosphorylated extracellular signal-regulated kinase; GRIP1: glucocorticoid receptor interacting protein 1; Hsp90: heat shock protein 90; NcoR1: nuclear receptor corepressor 1; P: phosphate; PBREM: phenobarbital responsive enhancer module; PP2A: protein phosphatase 2A; PXR: pregnane X receptor; PXRE: PXR-responsive element; RACK: receptor for activated C kinase 1; p-RACK: phosphorylated receptor for activated C kinase 1; RXR: retinoid X receptor; SMRT: silencing mediator of retinoid and thyroid receptors; SRC-1: steroid receptor coactivator 1; XRE: xenobiotic response element. Figure created with BioRender.com.

**Figure 3 cells-12-02752-f003:**
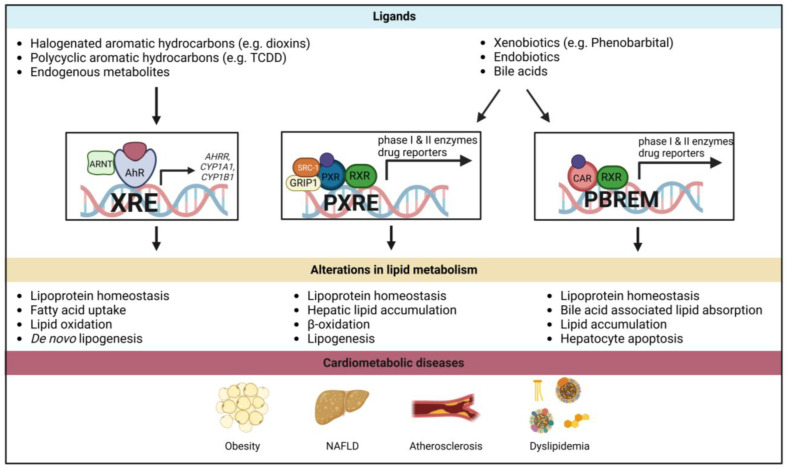
**Effects of xenobiotic receptor activation on lipid metabolism.** Activation of AhR (**left**) by ligands such as halogenated aromatic hydrocarbons leads to changes in gene transcription, which affects the lipid metabolism in various ways. PXR (**middle**) and CAR (**right**) share many ligands; however, their signaling pathways occur based on different cofactors. Nevertheless, both pathways result in similarly altered gene transcription of, for example, phase I and II enzymes. The activation of both receptors impacts lipid metabolism, targeting partly similar but also specific parts of lipid metabolism. Disruptions in lipid signaling can contribute to the development of CMDs, assigning the xenobiotic receptors a more metabolic sensing role. AhR: aryl hydrocarbon receptor; AHRR: aryl hydrocarbon receptor repressor; ARNT: aryl hydrocarbon receptor nuclear translocator; CAR: constitutive androstane receptor; CYP1A1: cytochrome P450 family 1 subfamily A member 1; CYP1B1: cytochrome P450 family 1 subfamily B member 1; GRIP1: glucocorticoid receptor interacting protein 1; NAFLD: non-alcoholic fatty liver disease; PBREM: phenobarbital-responsive enhancer module; PXR: pregnane X receptor; PXRE: PXR-responsive element; RXR: retinoid X receptor; SRC1: steroid receptor coactivator 1; TCDD: 2,3,7,8-tetrachlorodibenzo-p-dioxin; XRE: xenobiotic response element. Figure created with BioRender.com.

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
