# Peer review of "AhR, PXR and CAR: From Xenobiotic Receptors to Metabolic Sensors"

_cells, 2023, doi:10.3390/cells12232752_

Round 1

Reviewer 1 Report

Comments and Suggestions for Authors

The review article manuscript written by Rakateli et al. with the title AhR, PXR and CAR: from xenobiotic receptors to metabolic  sensors is well written with stress on mechanistic aspects of metabolism regulation.

The main weakness is that the manuscript is an analogy of another already published paper (Jingyuan Wang, Peipei Lu and Wen Xie. Atypical functions of xenobiotic receptors in lipid and glucose metabolism. Med. Rev. 2022; 2(6): 611–624).

Despite this, the manuscript still has some value.

Minor comments:

1.       Phenobarbitone, as well as phenytoin, are a common CAR/PXR activators. Interpretation of clinical results of some references (114-118) should be reconsidered and corrected (Wang et al. Expert Opin Drug Metab Toxicol. 2014 Nov; 10(11): 1521–1532.;  Kliewer et al. Endocrine Reviews, 23, 5, 2002, 687–702). CITCO is a CAR and a weak PXR ligand (Li et al. 2020)

2.       The review by Hakkola´s lab on PXR-mediated cholesterol modulation should be cited (by Karpale). Some recent reviews by H. Wang's laboratory should be cited. A paper by Oliver Burk on PXR-induced steatosis should be cited in ArchTox. Tasheng Chen's reviews should be also cited. A paper by Hoekstra et al., 2009 should be cited as well.

Author Response

Please see attached pdf file for the point-by-point reply.

Reviewer 2 Report

Comments and Suggestions for Authors

Hypertriglyceridemia, diabetes, stroke, and metabolic syndrome should be taken into consideration for discussion particular in clinic. 

All figures are self drawn, if some part is copied from other research article that needs to be gotten permission. 

Could the authors give one entire abbreviation list.

Author Response

(The authors gave the same response as above.)

Reviewer 3 Report

Comments and Suggestions for Authors

The review by Rakateli and collaborators is well presented. It summarizes and discusses current trends in the area of lipid metabolism (mainly) and nuclear receptors from the AhR, PXR and CAR family. The novelty is appropriate and the proposed comments are intended to implement and clarify a few points that, at this moment, are pertinent for readers of the review.

 Specific comments:

Figure 1 can be implemented by incorporating CD36 as an important scavenger receptor (in addition to SR-BI), since it is highly expressed in the liver and plays a role in the NASH/fibrosis progression.  

The primary report (2012) by Tanos et al in Hepatology should be included (doi: 10.1002/hep.25571), since it provides the experimental basis for the role of AhR on Cho biosynthesis and its use as a potential therapeutic target.

In my opinion, current trends from clinical trials targeting these nuclear receptors are very relevant. Indeed, they are improving our understanding of their role under pathological conditions of lipid dysregulation, from liver injury to cancers. For example, consider the work by Panzitt (10.1016/j.mce.2022.111678). A short list of current clinical trials in this field can be incorporated in a summarized way.

Minor:

Please include the LDLR definition in the upper symbol from fig. 1.

As a recent review on PXR in the metabolism of additional molecules to lipids, please consider to cite the work by Lv et al (doi: 10.3389/fendo.2022.959902).

Author Response

(The authors gave the same response as above.)
